# Self-Report versus Neuropsychological Tests for Examining Executive Functions in Youth Soccer Athletes—A Cross-Sectional Study

**DOI:** 10.3390/bs12090346

**Published:** 2022-09-19

**Authors:** Florian Heilmann

**Affiliations:** Movement Science Lab, Institute for Sport Science, Philosophical Faculty II, Martin-Luther University, Halle-Wittenberg, 06108 Halle (Saale), Germany; florian.heilmann@sport.uni-halle.de; Tel.: +49-345-5524454

**Keywords:** executive functions, self-reporting, cognitive diagnostics, talent diagnostics

## Abstract

Cognitive diagnostics, especially the measurement of executive functions (EFs) in the context of sports and talent diagnostics, is a popular research topic. However, research is lacking on how self-reports are sufficient to examine the EFs of youth athletes for performance diagnostics. Thus, the current study aims to evaluate the relationships between neuropsychological tasks (3-back task, cued Go/NoGo task, flanker task, and number-letter task) and a self-report for examining EFs (BRIEF-SB). Furthermore, it should be investigated whether it is possible to predict the outcome of EF tasks using a self-report inventory. Therefore, 68 young professional soccer players (M_age_ = 14.26 ± 1.35 years) from a national youth academy were included in the study. The weak-to-moderate correlations (*r* = 0.000, *p* = 0.999 to *r* = −0.442, *p* < 0.01) and the results of sensitivity analysis (0.125 to 0.538) do not support using a self-report of EFs for cognitive performance diagnostics. The inventory is only suitable for identifying executive dysfunctions in athletes recovering from head injuries or concussions.

## 1. Introduction

Executive functions (EFs) make it possible to act with purpose or in a goal-directed manner. According to Diamond [1], the definition of EFs includes the functions of working memory, inhibition, and cognitive flexibility. Expert performance in sports requires outstanding physical capabilities and motor control, perception, information processing, and cognitive functioning, such as EFs [2]. Previous studies have provided evidence that experts in sports have superior EFs compared to non-athletes (for review, see [3]). Furthermore, open-skill sports (i.e., nature and combat) tend to have the most significant positive influence on EFs (for review, see [4]). Sports such as soccer place high demands on the EFs because the athletes regularly have to adapt flexibly to situations (cognitive flexibility), stop actions at short notice (inhibition), or make situation-relevant information quickly available (working memory; [4]). There have been previous studies with different topics on EFs in youth sports, such as EFs in relation to physical performance [5] or EFs as diagnostic measures [6].

EF tests (i.e., flanker task, n-back task, and number-letter-task) can describe a subject’s cognitive performance. These neuropsychological examinations are an in-depth assessment of skills and abilities such as attention capacity, problem-solving, memory, IQ, or visual-spatial skills that are linked to specific brain functions.

The authors of some studies [7,8,9] have suggested implementing the measurement of EFs as a cognitive part of performance diagnostic measures. Furthermore, authors have critically discussed neuropsychological diagnostics [10]. For instance, Beavan et al. [11,12] described the age-dependent function of EF development. The development is oriented to general populations reported in previous research, which means that it is difficult to differentiate and delineate different development processes between individuals. The age of participants is a relevant factor in every analysis dealing with adolescents because there are substantial changes in EF performance at this age [13]. Thus, it is questionable using EFs to describe the cognitive skills of athletes. Nevertheless, it is established that athletes should have superior cognitive skills [3], and impairments could significantly influence their performance in sports [14]. In contact sports (i.e., ice hockey or football), EFs could be a predictor of having a concussion [15]. The EF performance of players recovering from a concussion is impaired [14].

The current study attempts to retrieve the idea of Beavan et al. [11] to implement the threshold hypothesis in consideration of EFs in sports. The theory describes that only a critical value regarding the domain-generic cognitive functions must be reached to achieve high performance in a team sport [11]. However, knowledge is lacking on whether this critical value must be examined using computerized neuropsychological tests or whether a self-report of EFs would be suitable to identify the athletes with low domain-generic cognitive performance.

EFs could be assessed by parents, educators, coaches, as well as independent qualified observers or through a self-report. An example of self-reporting EF inventory is the Behavior Rating Inventory of Executive Function (BRIEF-SB; [16]). The inventory is a clinical scale to examine and interpret executive dysfunctions in a daily life setting. For example, the rating has been used in previous studies to identify the executive dysfunctions of athletes after a history of concussion [17] or, in general, to examine the neuropsychiatric and cognitive outcomes of contact sports athletes [17]. In contrast to the EFs, a dysfunction is referred to as a functional limitation that has an impact on everyday life or, as in the current case, on performance in a particular sport. The authors explicitly point out that the BRIEF scales are based on the circumscribed neuropsychological functions, but that no conclusions can be drawn about neuroanatomical substrates [18]. Nevertheless, the inventory results could allow deductions about EFs or their falling below a critical value to achieve high performance in the relevant sport.

Knowledge is lacking about using a self-reporting inventory to assess the EF of youth athletes. Two studies have reported the validity of self-reports and compared it to neuropsychological task assessments. For instance, Buchanan [19] showed that the self-report of EFs using the WebEXEC tool [20] did not correlate with performance on Trail Making, Phonemic Fluency, Semantic Fluency, or Digit Span tests tapping executive function in non-clinical samples. The findings of the study by Bryant et al. [21] were that subjective cognition is not significantly associated with any objective measures of cognitive functioning in adults. 

The current exploratory study aims to examine the correlations between the commonly used neuropsychological tasks (3-back task, cued Go/NoGo task, flanker task, and number-letter task) and scores of BRIEF-SB to assess EFS in youth athletes. The investigation should clarify the extent to which the inventory can predict poor EF performance demonstrated by neuropsychological testing. 

It is hypothesized that the self-reported EFs correlate with the results of the neuropsychological measurements. Therefore, subscales and specific tasks representing the same construct of EFs (i.e., inhibition) should show the highest correlations. 

## 2. Materials and Methods

### 2.1. Participants

Sixty-eight male young players of a national youth soccer academy (highest German youth league) were included in the study (*R*_age_ = 11–16 years, *M*_age_ = 14.26 years, *SD* = 1.35 years). The average training age (mean years of experience playing soccer in a structured academy) was *M*_tage_ = 9.12 years (*SD* = 2.51 years). The inclusion criteria were an affiliation with the youth academy and frequent training and physical health. Exclusion criteria were uncorrected ametropia or a history of concussion in the last six months.

All procedures performed in the studies involving human participants adhered to the ethical standards of the institutional research ethics committee and the 1964 Helsinki Declaration and its later amendments or comparable ethical standards. The procedures are a standard assessment in half-yearly diagnostic tests. Informed consent was obtained from all participants or legal representatives included in the study.

### 2.2. Neuropsychological EF Tasks

Computerized neuropsychological tests with Inquisit Lab 6 (Millisecond Software LLC, Seattle, WA, USA) were performed to describe EFs on a 17-inch screen and a QWERTZ keyboard. An Eriksen-flanker task [22] and cued Go/NoGo task [23] were utilized to assess participants’ inhibition. The Eriksen-flanker task was chosen to evaluate the ability of the cognitive function to fade out distractions, and the Go/No go task was used to evaluate the ability to suppress a simple reaction. Furthermore, a 3-back task [24] was used to examine participants’ working memory. To evaluate cognitive flexibility, a number-letter task was carried out that was modified from the Alternating-Runs-Switch task by [25]. For further description of EF tasks, see [26]. The tasks have been validated and used in several studies to assess EFs the manners described above [1].

### 2.3. Self-Reporting of EFs

The German version of the BRIEF-SB scale was used for the self-reporting of EFs [18,27]. The scale is an inventory of 80 items representing eight index scales (Inhibit, Shift, Emotional Control, Monitor, Working Memory, Plan/Organise, Organisation of Materials, and Initiate) and two validity scales (Inconsistency and Negativity). Items such as “If I’m given three orders at once, I can only remember the first or third” have to be answered with a three-point Likert scale: “never/very rarely” (1), “sometimes” (2), or “often” (3). Total scores of the “behavior regulation index” (BRI), “cognitive regulation index” (CRI), and an overall score (TS) of the eight index scores were calculated [18]. The scale has a well-established construct and predictive validity and internal consistency of Cronbach’s α between 0.73–0.85 for index items and 0.96 for total score [18].

### 2.4. Procedures

The study was a survey (study type: descriptive cross-sectional) among professional youth soccer players. The soccer players were tested at their training facilities from September 2021 to November 2021. First, players arrived at the facilities and were greeted by the experimenter (author). Next, they were educated about the procedures and requested to sign the informed consent. After that, the players completed the BRIEF-SB inventory and performed the cognitive tasks, which lasted approximately 45 min. The order of neuropsychological tasks switched in randomized order to control for sequence effects. The players were tested in a quiet room one hour before training to avoid physical exercise effects, between 10:00 a.m. and 4:00 p.m.

### 2.5. Statistical Analysis

The Bravais–Pearson correlation was calculated between neuropsychological EF tasks’ parameters and BRIEF-SB items. The fulfillment of requirements for parametric correlation analysis was checked. Correlations of ±1 are specified as perfect, ±0.70–0.99 as strong, 0.40–0.69 as moderate, and 0.01–0.39 as weak [28]. Discriminatory power was calculated with a 2 × 2 contingency table for the total score of BRIEF-SB. The value specifies the ability to identify the 20 worst values of neuropsychological tasks by using the BRIEF-SB inventory (sensitivity = number of true positives/number of true positives + number of false negatives). The number of 20 worst scores was set because it would be interesting for practitioners to identify the worst 20 players in terms of EFs from a sample of 68 players. A sensitivity of 0.75 is acceptable in this study. Statistical analyses were performed using SPSS 28 (SPSS, Chicago, IL, USA). A significance level of 0.05 was chosen.

## 3. Results

### 3.1. Descriptive Results

The descriptive results of performance-based EF tasks (neuropsychological tasks) are displayed in Table 1, and the results of self-report (BRIEF-SB) are shown in Table 2.

### 3.2. Correlational Analysis

The results of the correlational analysis show significant negative correlations between the accuracy of the 3-back task for evaluating working memory and the index “shift” (*r*(63) = −0.251, *p* = 0.045) and “plan/organise” (*r*(63) = −0.276, *p* = 0.028).

Furthermore, significant correlations could be identified between the indexes “monitor”, “working memory”, and “plan/organise” and the parameters of response time in the cued Go/NoGo task (mean, vertical, and horizontal cue; *r*(67) = 0.253, *p* = 0.037 to *r*(67) = 0.371, *p* = 0.002), representing the inhibition of participants. There were positive significant relationships between total scores (BRI, CRI, and TS) and response times in the cued Go/NoGo task (*r*(67) = 0.251, *p* = 0.039 to *r*(67) = 0.288, *p* = 0.017).

No significant correlation could be identified between the BRIEF-SB scores and the parameters of the flanker task (inhibition).

The correlations between index scales of BRIEF-SB and parameters of the number-letter task could be quantified between *r*(59) = −0.254, *p* = 0.050 and *r*(59) = −0.442, *p* < 0.01 and *r*(59) = −0.260, *p* = 0.045 and *r*(59) = −0.359, *p* = 0.005 for total scores (mainly between the indices “Emotional Control”, “Monitor” and accuracy parameters of the number-letter task). The entire results can be accessed in the Appendix A. The calculation of partial correlations by taking into consideration the age of participants shows no different results.

### 3.3. Sensitivity Analysis

The sensitivity analysis results in values between 0.125 and 0.538 to identify low performance in neuropsychological tasks by using the total score of self-reported EFs (BRIEF-SB inventory).

## 4. Discussion

This study aimed to examine the correlations between self-reported EFs and the results of computerized neuropsychological tests. Furthermore, the findings attempted to outline whether the results of the BRIEF-SB inventory allow the outcomes of the EF tests to be predicted (for example, the ability to identify the participants with bad EF performance).

The results of the exploratory study show only weak-to-moderate correlations between the parameters of EF tests and self-reporting. Even if the studies of Buchanan [19] and Bryant et al. [21] do not report findings regarding participants with the same characteristics, it could be stated that the studies show similar results. They also provide evidence that self-reports could not substitute neuropsychological measurements of EFs. In the current study, the self-report inventory (BRIEF-SB) is a valid instrument concerning participants’ age. Still, the questionnaire asks for specific behavioral issues, trying to infer back to the EFs.

In most cases, athletes show no relevant issues concerning their EFs. Moreover, some studies show that it is difficult to differentiate between players in homogenous groups with high EF performance. The study shows that distinguishing between subjects is even more challenging when the examination method is not appropriate.

However, the use of the BRIEF inventory is much more economical with regard to the time, costs, and motivation spent on the neuropsychological diagnostics in soccer teams or youth academies (i.e., maybe teams cannot afford the computer tests in the first place). Nyongesa et al. [29] reported in a scoping review that performance-based measurements (neuropsychological tasks) are frequently used in adolescent EF assessment. They conclude that this has to be based on the fact that there are only a few rating scales or self-reports, such as the BRIEF inventory. Nevertheless, in the present study, the self-report could not be described as a suitable tool to examine the EFs of athletes and the deriving of performance-specific insights. The current findings do not support the use of self-reports of EFs to assess the cognitive functions of athletes, and a critical value could not be established.

However, using a self-report is conceivable when dysfunctions from trauma or lesions of specific brain structures are expected and when one wants to obtain information about the cognitive functions of an athlete [14,17].

Furthermore, the sensitivity analysis results do not reveal sensitivity values above a critical level of 0.75. The low scores signify that the self-report inventory of EFs could not assess domain-generic cognitive skills or predict bad results in neuropsychological EF tasks. However, neuropsychological tasks are suggested to determine EFs in athletes because practitioners can identify dysfunctions using these measurements, allowing discrimination between superior and poor EF performance.

## 5. Limitations

Three limitations inherent in the current study must be considered when interpreting the results. First, the items of the BRIEF-SB inventory are often referred to in a school setting. When assessing the EFs of athletes, an inventory referring to a sports context would be more appropriate. Secondly, a response scheme that follows socially desirable responses cannot be ruled out for the athletes. A two-sided approach of a self-report and possibly a parental report is suggested for future studies. Thirdly, the correlation between the self-report and neuropsychological tests could be examined in this current study. A critical value could not be identified, which possibly could have been uncovered by a study with an expert–novice paradigm similar to what Sakamoto et al. [7] did in their study. The athletes examined in the present study were retrieved from an elite academy in Germany. The study demonstrates the EF profiles of a homogenous group of athletes who are already enrolled in the academy, so clearly, they have an adequate performance level to compete.

## 6. Conclusions

Despite the current study’s limitations, one could conclude from the results that a self-report of EFs is unsuitable for evaluating the EF performance for cognitive skills diagnostics or similar intentions. Instead, it is applicable only for identifying conspicuous issues in the cognitive function of athletes or athletes with known head injuries. Future research in this field should focus on developing and validating specific inventory to examine EFs in athletes. The recently available inventories do not solve the problem and neglect the special characteristics of athletes.

## Figures and Tables

**Table 1 behavsci-12-00346-t001:** Descriptive results of performance-based EF tasks.

Task	Parameter	Mean	*SD*	Range	
				Min	Max
Cued Go/NoGo task	error rate (%)	0.02	0.06	0.00	0.50
	mean RT (ms)	315.16	31.89	269.88	420.40
	mean RT horizontal cue (ms)	312.11	30.96	262.46	412.00
	mean RT vertical cue (ms)	327.43	40.54	222.28	458.30
Flanker task	% of correct answers	0.97	0.03	0.78	1.00
	mean RT (ms)	446.80	62.93	326.94	629.38
	mean RT congruent (ms)	436.83	62.71	323.63	614.13
	mean RT incongruent (ms)	468.42	70.10	333.29	742.90
3-back task	mean RT (ms)	764.36	276.20	360.11	1673.30
	accuracy (%)	14.13	5.87	0.00	26.08
Number-letter task	% correct answers switch trials (ms)	0.86	0.11	0.53	1.00
	% correct answers non-switch trials	0.93	0.09	0.50	1.00
	accuracy (%)	−0.07	0.09	−0.35	0.15
	mean RT switch trials (ms)	1518.37	426.17	783.17	2627.29
	mean RT non-switch trials (ms)	1014.91	297.23	555.00	2068.52
	mean RT switch costs (ms)	503.46	296.55	−200.16	1300.98

RT = response time.

**Table 2 behavsci-12-00346-t002:** Descriptive results of BRIEF-SB inventory.

	Inhibit	Shift	Emotional Control	Monitor	Working Memory	Plan	Organize	Initiate	Total Score
**mean**	19.34	15.72	14.26	7.31	18.01	20.68	10.44	15.16	120.93
* **SD** *	3.16	2.94	2.67	1.71	3.69	3.75	2.66	3.03	17.00
**min**	14.00	10.00	10.00	5.00	12.00	13.00	7.00	10.00	82.00
**max**	31.00	22.00	23.00	12.00	28.00	28.00	18.00	22.00	170.00

## Data Availability

The data presented in this study are available on request from the corresponding author.

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
