# Peer review of "Self-Report versus Neuropsychological Tests for Examining Executive Functions in Youth Soccer Athletes—A Cross-Sectional Study"

_behavsci, 2022, doi:10.3390/bs12090346_

Round 1

Reviewer 1 Report

First of all, thank you for the opportunity to review this exciting manuscript titled: ”Self-Report versus Neuropsychological tests for Examining Executive Functions in Youth Soccer Athletes.

There are a few comments and recommendations listed below:

Introduction

In the first paragraph, the definition of executive functions is fragile. I would recommend citing the definition from cognitive psychology. Moreover, in lines 27 to 30, the author writes: “Previous studies provide evidence that experts in 27 sports have superior EFs compared to non-athletes (for review, see [3]. Furthermore, open-skill sports (i.e., nature and combat) tend to have the most significant positive influence on EFs (for review, see [4])”. There should be more explanation on that matter. Why do combat sports influence the EFs the most? How about team sports and soccer in particular? What is the level of knowledge and research on that aspect?

In the second paragraph, I recommend developing the information on the state of research on executive functions in youth sports. There is also a lack of definition for the term neuropsychological.

In lines 44-45, the author mentions that only neuropsychological tests or self-reports could be used to estimate the level of executive functions. However, we also have a massive battery of psychomotor tests, such as the Vienna Test System. So I would recommend adding the statement on psychomotor tests as well as neuropsychological tests or self-reports.

In line 48, the author writes that parents and educators could assess EFs. There should also be a piece of information that EFs could be assessed by coaches as well as independent qualified observers.

Material and Methods

What was the gender of the players?

Neuropsychological EF Tasks and each test should be described separately and more accurately.

In the section Statistical Analysis, in lines 108-109, the author specifies the correlation intervals. How do the authors specify that with 68 participants, such intervals (strong, moderate, weak) are appropriate?

Discussion

In line 152, there should be cannot instead of can’t.

The discussion could have the structure of two paragraphs in general—the first paragraphs on the results lines 142 to 165—the second paragraph on limitations lines 166 to 178.

Conclusions

The author should avoid the term talent diagnosis because this aspect is controversial in sports science and practice. I would recommend using “cognitive skills diagnosis.”

Author Response

[Comment 1]:

In the first paragraph, the definition of executive functions is fragile. I would recommend citing the definition from cognitive psychology.

[Response]:

Thank you very much for your comment on this definition. I was a bit irritated about the statement that the definition is fragile because this definition of Diamond (2012) is established in psychology and cited over 11000 times in research.

[Comment 2]:

Moreover, in lines 27 to 30, the author writes: "Previous studies provide evidence that experts in 27 sports have superior EFs compared to non-athletes (for review, see [3]. Furthermore, open-skill sports (i.e., nature and combat) tend to have the most significant positive influence on EFs (for review, see [4])". There should be more explanation on that matter. Why do combat sports influence the EFs the most? How about team sports and soccer in particular? What is the level of knowledge and research on that aspect?

[Response]:

Thank you very much. I agree with the comment that I should explain the benefits of open-skill sports in more detail. I added the following sentence to consider the three EFs: "Sports such as soccer place high demands on the EFs because the athletes regularly have to adapt flexibly to situations (cognitive flexibility), stop actions at short notice (inhibition), or make situation-relevant information quickly available (working memory; [4]). "(lines 30-33)

[Comment 3]:

In the second paragraph, I recommend developing the information on the state of research on executive functions in youth sports.

[Response]:

Thank you very much for the comment. I agree that I could have provided more information on EFs in youth sports, especially youth soccer. I added the following sentence: "There are studies with different topics also on EFs in youth sports: relation to physical performance [REF], EFs as diagnostic measures [REF]." (lines 32-34)

REF:

Verburgh, L.; Königs, M.; Scherder, E.J.A.; Oosterlaan, J. Physical exercise and executive functions in preadolescent children, adolescents and young adults: a meta-analysis. Br. J. Sports Med. 2014, 48, 973–979, doi:10.1136/bjsports-2012-091441.

Beavan, A.; Spielmann, J.; Ehmann, P.; Mayer, J. The Development of Executive Functions in High-Level Female Soccer Players. Percept. Mot. Skills 2022, 129, 1036–1052, doi:10.1177/00315125221096989.

[Comment 4]:

There is also a lack of definition for the term neuropsychological.

[Response]:

I agree with this, and I added a definition of neuropsychological testing because it describes the testing and is more specific in this part of the manuscript than only the description of "neuropsychological". The following definition was added: "These neuropsychological examinations are an in-depth assessment of skills and abilities such as attention capacity, problem-solving, memory, I.Q. or visual-spatial skills linked to specific brain functions. "(lines 35-37)

[Comment 5]:

In lines 44-45, the author mentions that only neuropsychological tests or self-reports could be used to estimate the level of executive functions. However, we also have a massive battery of psychomotor tests, such as the Vienna Test System. So I would recommend adding the statement on psychomotor and neuropsychological tests or self-reports.

[Response]:

Thank you very much for the comment. I think this statement could be based on a misunderstanding. The tasks we used are no psychomotor tasks, such as the pegboard test or the pursuit tracking task, and psychomotor tasks could not be used to describe EFs. Instead, we examined EFs by neuropsychological tasks.

[Comment 6]:

In line 48, the author writes that parents and educators could assess EFs. There should also be a piece of information that EFs could be assessed by coaches as well as independent qualified observers.

[Response]:

Thank you very much for your suggestion. I added these two possibilities in the text. (lines 56-57)

Material and Methods

[Comment 7]:

What was the gender of the players?

[Response]:

Thank you very much for reading the paper in that precise manner. I forgot to mention that we only tested male soccer players because there are no female teams with this sample size at the youth academy.

[Comment 8]:

Neuropsychological EF Tasks and each test should be described separately and more accurately.

[Response]:

Thank you very much for your comment. The form of the manuscript is defined as Brief Report, so I tried to cite the relevant literature, and a detailed description is available from the last cited paper: "For further description of EF tasks, see [20]. "(lines 86-87) I hope that is suitable for the Brief Report.

[Comment 9]:

In the section Statistical Analysis, in lines 108-109, the author specifies the correlation intervals. How do the authors specify that with 68 participants, such intervals (strong, moderate, weak) are appropriate?

[Response]:

Thank you for the comment. The intervals are established and I cited the relevant literature. They do not differ in sample sizes as in my study.

Discussion

[Comment 10]:

In line 152, there should be cannot instead of can't.

[Response]:

Thank you again for reading the manuscript so carefully. I revised the word.

[Comment 11]:

The discussion could have the structure of two paragraphs in general—the first paragraphs on the results lines 142 to 165—the second paragraph on limitations lines 166 to 178.

[Response]:

Thank you very much for the comment. To address this suggestion, I split the discussion section into Discussion (lines 144-168) and Limitations (lines 171-183).

Conclusions

[Comment 12]:

The author should avoid the term talent diagnosis because this aspect is controversial in sports science and practice. I would recommend using "cognitive skills diagnosis."

[Response]:

I agree with that. I revised this controversial wording: "Despite the current study's limitations, one could conclude that a self-report of EFs is unsuitable for evaluating the EF performance for cognitive skills diagnostics or similar intentions. "(lines 185-187)

Reviewer 2 Report

Thank you for the opportunity to review the manuscript „Self-Report versus Neuropsychological Tests for Examining Executive Functions in Youth Soccer Athletes”. I believe this work has good potential, but I think it could benefit if the Authors address some major issues mentioned below.

1. The Introduction

The Introduction section should be more detailed and comprehensive. The main constructs are quite poorly elaborated. Please provide more detailed descriptions of the main constructs. The Introduction lacks a specific theoretical framework. Please, state succinctly and clearly the hypotheses (including the specific EFs subtest and subscales). 

Please, be precise: The sentence: "The EF tests (i.e., flanker task, n-back task, and trail-making task) can describe the 31 cognitive performance of athletes" (p. 1, lines 31-32) seems to suggest that those measures are specific for athletes (what is not true).

Please rewrite the sentences: "For instance, [9, 10] described the age-dependent function of EF development. This function is oriented towards the development of general populations reported in previous research." (p. 1, lines 35-36) as it is unclear.

Please, substantiate the statement: "In contact sports 39 (i.e., ice hockey or football) EFs could be a predictor of having a concussion [12]." (p. 1, lines 39-40).

The Introduction does not include the developmental context of EFs in adolescence. Meanwhile, it is a time of substantial changes in EFs.

2. Materials and Methods

This section lacks some crucial details as follows:

What was the exact range of participant's age? Why age was not controlled for in the analyses, regarding the fact that adolescence is a time of rapid EF growth?

What was the SES of the participants?

Was the study approved by an ethics committee?

Please describe all the performance measures of EFs, including their exact indices. Why inhibitory control was measured with 3 measures (as opposed to the remaining EFs)?

3. Results section

Please provide a table with descriptive statistics for all the continuous variables (M, SD, range, skewness, kurtosis).

Please, provide a table with correlation analyses in the manuscript (instead of quoting all the data in the text). What was the range of age? Why age was not controlled in the analyses (regarding the fact that adolescence is a time of rapid EF growth)?

Section 3.1. Sensitivity Analysis seems incomplete. Please provide the full description of the analysis (and the tables).

4. Discussion

The Discussion section strongly needs to be improved, especially in terms of its insightfulness:

The Author states: "The findings could not be compared with other studies because, to our knowledge, no investigations relate the two instruments of evaluating EFs." (p. 4, lines 147-148). That's not true - see e.g., https://pubmed.ncbi.nlm.nih.gov/26191609/

On the p. 4, lines. 153-154 we read: "Nevertheless, in the present study, the self-report could not be described as a suitable tool to examine the EFs of athletes and the deriving of performance-specific insights." - please develop and justify this thesis in detail (including also the state of the art).

It would be worth discussing the negative correlations between the measures of EFs, obtained in the Author's study ("The results of correlational analysis show significant negative correlations between 120 the accuracy of 3-back task for evaluating working memory and the index "shift" (r[63] = 121 -.251, p = .045) and "plan / organise" (r[63] = -.276, p = .028)." (p. 3)

Please discuss your results in relation to the issues of validity—functionality, ecological validity, and ethological validity of EFs measures. Moreover, the performance-based EFs measures used in this study (but also in general) tend to be crude and underspecified in terms of the cognitive processes that they engage in - please refer to this problem while discussing the results. 

On page 4 Author states: "The study demonstrates the EF profiles of a homogenous group of athletes who are al-176 ready enrolled in the academy, so; clearly, they have an adequate performance to compete 177 (including their EF profile)." - please, refer to the basis on which this conclusion was drawn? Did the Author have any psychometric norms? For which measures?

I suggest discussing your results in the context of the problem of assessing EFs in adolescence - see e.g., https://www.frontiersin.org/articles/10.3389/fpsyg.2019.00311/full, https://www.sciencedirect.com/science/article/pii/S0887617707001928

Much more attention should be paid to the limitations of the study. 

The manuscript needs proofreading.

Author Response

Thank you for the opportunity to review the manuscript "Self-Report versus Neuropsychological Tests for Examining Executive Functions in Youth Soccer Athletes". I believe this work has good potential, but I think it could benefit if the Authors address some major issues mentioned below.

[Comment 1]:

The Introduction section should be more detailed and comprehensive. The main constructs are quite poorly elaborated. Please provide more detailed descriptions of the main constructs. The Introduction lacks a specific theoretical framework. Please, state succinctly and clearly the hypotheses (including the specific EFs subtest and subscales).

[Response]:

Thank you very much for the comment. I agree with the statement, and I added two paragraphs to firstly elaborate on the relevant constructs and secondly refer to studies that have similar research questions:

"Sports such as soccer place high demands on the EFs because the athletes regularly have to adapt flexibly to situations (cognitive flexibility), stop actions at short notice (inhibition), or make situation-relevant information quickly available (working memory; [4]). There are studies with different topics also on EFs in youth sports: relation to physical performance [REF] or EFs as diagnostic measures [REF]"(lines 31-36)

"Two studies report the validity of self-reports and compare it to neuropsychological task assessments. For instance, REF [] showed that the self-report of EFs using the WebEXEC tool [REF] did not correlate with performance on Trail Making, Phonemic Fluency, Semantic Fluency, or Digit Span tests tapping executive function in non-clinical samples. The second study's findings of REF [] were that subjective cognition is not significantly associated with any objective measures of cognitive functioning in adults. "(lines 72-77)

Regarding the hypothesis, I added the following sentence. But I have to remind that the study is exploratory:

"It is hypothesized that the self-reported EFs correlate with the results of the neuro-psychological measurements. Subscales and specific tasks representing the same construct of EFs (i. e., inhibition) should show the highest correlations. "(lines 83-85)

[Comment 2]:

Please, be precise: The sentence: "The EF tests (i.e., flanker task, n-back task, and trail-making task) can describe the 31 cognitive performance of athletes" (p. 1, lines 31-32) seems to suggest that those measures are specific for athletes (what is not true).

[Response]:

Thank you for the comment. I removed the end of the sentence, and now it should be clear that neuropsychological tests could examine EF in humans.

[Comment 3]:

Please rewrite the sentences: "For instance, [9, 10] described the age-dependent function of EF development. This function is oriented towards the development of general populations reported in previous research." (p. 1, lines 35-36) as it is unclear.

[Response]:

Thank you very much for the comment and for reading the paper in that precise manner. I revised the sentence as follows:

"The development is oriented to general populations reported in previous research, which means that it is difficult to differentiate and delineate different development processes between individuals. "(lines 44-48)

[Comment 4]:

Please, substantiate the statement: "In contact sports 39 (i.e., ice hockey or football) EFs could be a predictor of having a concussion [12]." (p. 1, lines 39-40).

[Response]:

Thank you very much. I also think this sentence could have a short explanation. I added the following sentence to underpin the relevance of EFs in concussion history or rehabilitation: "The EF performance of players recovering from a concussion is impaired [12]. "(lines 50-51)

[Comment 5]:

The Introduction does not include the developmental context of EFs in adolescence. Meanwhile, it is a time of substantial changes in EFs.

[Response]:

You are right. I mentioned it now in the introduction. The developmental context is relevant but to a minor extent in this paper because we only check if the self-report is suitable to differentiate individuals in this sample. Of course, the sample is at a relevant age for growth spurts, etc.:

"The age of participants is a relevant factor in every analysis dealing with adolescents because there are substantial changes in EF performance in this age [REF]. "(lines 48-50)

[Comment 6]:

What was the exact range of the participant's age? Why age was not controlled for in the analyses, regarding the fact that adolescence is a time of rapid EF growth?

[Response]:

Thank you very much for the questions. I added the exact range of age in the materials and methods section and added a sentence concerning the integration of age in the calculation of correlation:

"The calculation of partial correlations by taking into the age of participants show no different results. "(lines 160-161)

[Comment 7]:

What was the SES of the participants?

[Response]:

Thank you very much for the question. I clarified that the participants were male in the materials and methods section.

[Comment 8]:

Was the study approved by an ethics committee?

[Response]:

Yes, the study was approved, and the editorial office got all information and the approval code. Therefore, I revised the section to give more details:

"All procedures performed in the studies involving human participants adhered to the ethical standards of the institutional research ethics committee e and the 1964 Helsinki Declaration and its later amendments or comparable ethical standards. The procedures are a standard assessment in half-yearly diagnostic tests. Informed consent was obtained from all participants or legal representatives in the study. "(lines 95-99)

[Comment 9]:

Please describe all the performance measures of EFs, including their exact indices.

[Response]:

Thank you very much for the comment. I agree with the statement that the measures have to be described in detail. However, the article type is Brief Report, so I hope it is acceptable for you that I cited a paper where the tasks are described comprehensively:

"For further description of EF tasks, see [20]. The tasks are validated and used in several studies to assess EFs described above [1]. "

[Comment 10]:

Why inhibitory control was measured with 3 measures (as opposed to the remaining EFs)?

[Response]:

Thank you very much for reading the paper in that precise manner. I used two tasks to describe inhibition of participants. I added a sentence in materials and methods to explain why I did this:

"The Eriksen-flanker task was chosen to evaluate the cognitive function to fade out distractions and Go/No go task to suppress a simple reaction. "(lines 104-106)

[Comment 11]:

Please provide a table with descriptive statistics for all the continuous variables (M, SD, range, skewness, kurtosis).

[Response]:

Thank you very much for this comment. I agree with the statement that the descriptive values have to be presented in the manuscript. Therefore, I added two tables with the data of EF tasks (performance-based) and the data of the self-report.

[Comment 12]:

Please, provide a table with correlation analyses in the manuscript (instead of quoting all the data in the text).

[Response]:

Thank you for this comment. The correlational analysis with 16 variables of performance-based EF assessment and 15 items of the BRIEF scale as a cross table is too large to present in the text. I added a supplementary file with the cross table and two tables with the descriptive data in the text.

[Comment 12]:

What was the range of age? Why age was not controlled in the analyses (regarding the fact that adolescence is a time of rapid EF growth)?

[Response]: Thank you for the comment. I agree. I added the range of the participant's ages. I did a partial analysis with age, but no different results exist. I reported this as follows:

"The calculation of partial correlations by taking into the age of participants show no different results. "(lines170-171)

[Comment 13]:

Section 3.1. Sensitivity Analysis seems incomplete. Please provide the full description of the analysis (and the tables).

[Response]:

Thank you very much for your detailed comments. The analysis is complete, and it is only explorative. I revised the sentence to clarify that the range of the two values allows concluding that the sensitivity is too low to identify the participants with bad performance-based EF values by the self-report.

"The sensitivity analysis results in values between 0.125 and 0.538 to identify low performance in neuropsychological tasks by using the total score of self-reported EFs (BRIEF-SB inventory). "(lines 173-175)

[Comment 14]:

The Author states: "The findings could not be compared with other studies because, to our knowledge, no investigations relate the two instruments of evaluating EFs." (p. 4, lines 147-148). That's not true - see e.g., https://pubmed.ncbi.nlm.nih.gov/26191609/

[Response]:

Thank you very much for this comment and the suggested reference. I added a paragraph for a specific state of research in the introduction and a discussion on that in the discussion section.

"Two studies report the validity of self-reports and compare it to neuropsychological task assessments. For instance, REF [] showed that the self-report of EFs using the WebEXEC tool [REF] did not correlate with performance on Trail Making, Phonemic Fluency, Se-mantic Fluency, or Digit Span tests tapping executive function in non-clinical samples. The second study's findings of REF [] were that subjective cognition is not significantly associated with any objective measures of cognitive functioning in adults. "(lines 73-78)

"Even if the studies of REF [] and REF [] do not report findings regarding participants with the same characteristics, it could be stated that the studies show similar results. They also provide evidence that self-reports could not substitute neuropsychological measurements of EFs. In the current study, the self-report inventory (BRIEF-SB) is a valid instrument concerning participants' age. Still, the questionnaire asks for specific behavioral issues, trying to infer back to the EFs. "(lines 174-179)

[Comment 15]:

On the p. 4, lines. 153-154 we read: "Nevertheless, in the present study, the self-report could not be described as a suitable tool to examine the EFs of athletes and the deriving of performance-specific insights." - please develop and justify this thesis in detail (including also the state of the art).

[Response]:

Thank you very much for the suggestion. I hope the previous revised discussion makes this statement clearer to the reader.

[Comment 16]:

It would be worth discussing the negative correlations between the measures of EFs, obtained in the Author's study ("The results of correlational analysis show significant negative correlations between 120 the accuracy of 3-back task for evaluating working memory and the index "shift" (r[63] = 121 -.251, p = .045) and "plan / organise" (r[63] = -.276, p = .028)." (p. 3)

[Response]:

Thank you very much. In my opinion, the correlations are discussed precisely. The values characterize weak to moderate correlations. The direction, in this case, could be negative because of the direction of interpretation. A high value in accuracy describes a good performance in EF tasks, and a low value in subscale shift represents good EFs. However, the correlation is only low to moderate, and sensitivity shows that we could not even identify the worst 20 participants with low performance-based EFs.

[Comment 17]:

Please discuss your results in relation to the issues of validity—functionality, ecological validity, and etiological validity of EFs measures. Moreover, the performance-based EFs measures used in this study (but also in general) tend to be crude and underspecified in terms of the cognitive processes that they engage in - please refer to this problem while discussing the results.

[Response]:

Thank you very much for the suggestion. I added the following paragraphs to the manuscript:

"In most cases, athletes show no relevant issues concerning their EFs. Moreover, some studies show that it is difficult to differentiate between players in homogenous groups with high EF performance. The study shows that distinguishing between subjects is even more challenging when the examination method is not appropriate.

However, using the BRIEF inventory is much more economical regarding the time, costs, and motivation spent on the neuropsychological diagnostics in soccer teams or youth academies (i.e., maybe teams can't afford the computer tests in the first place). REF [] reported in a scoping review that performance-based measurements (neuropsychological tasks) are frequently used in adolescent EF assessment. They conclude that this has to be based on the fact that there are only a few rating scales or self-reports, such as the BRIEF inventory. "(lines 180-190)

[Comment 18]:

On page 4 Author states: "The study demonstrates the EF profiles of a homogenous group of athletes who are al-176 ready enrolled in the academy, so; clearly, they have an adequate performance to compete 177 (including their EF profile)." - please, refer to the basis on which this conclusion was drawn? Did the Author have any psychometric norms? For which measures?

[Response]:

Thank you very much for this comment. I think the sentence is a misunderstanding. I revised it to clarify that we did not compare the profiles with norms.

[Comment 19]:

I suggest discussing your results in the context of the problem of assessing EFs in adolescence - see e.g., https://www.frontiersin.org/articles/10.3389/fpsyg.2019.00311/full

[Response]:

Thank you very much for the suggestion. I added a sentence in the discussion section and cited the suggested paper:

"REF [] reported in a scoping review that performance-based measurements (neuropsychological tasks) are frequently used in adolescent EF assessment. They conclude that this has to be based on the fact that there are only a few rating scales or self-reports, such as the BRIEF inventory. "(lines 185-188)

REF:

https://www.frontiersin.org/articles/10.3389/fpsyg.2019.00311/full

[Comment 20]:

The manuscript needs proofreading.

[Response]:

Thank you for the comment. A native speaker proofed the manuscript after the revision.

Reviewer 3 Report

STRUCTURE

-       The manuscript is properly structured.

TITLE AND ABSTRACT

- The Abstract is properly structured, but according to the magazine's instructions (https://www.mdpi.com/journal/behavsci/instructions), what was the methodology used? (change the structure). For example, “68 young professional football players (Mage = 14.26 ± 1.35 years) from a national youth academy were included in the study. Relationships between neuropsychological tasks (3-back task, cue GoNoGo task, flanker task and number-letter task) and a self-report for examining EFs (BRIEF-SB)”.

- The title or abstract should inform that the type of study.

- Line 9: words are shown in red and crossed out, check sentence. Applicable to the rest of the document.

INTRODUCTION

-       Line 31:The EF tests (i.e., flanker task, n-back task, and trail-making task) can describe the cognitive performance of athletes”. It is recommended to add this sentence at the end of the paragraph. Remove "in conclusion".

-       The following tests are discussed in the introduction: flanker task, n-back task, trail-making task, 3-back task, cued GoNoGo task and number-letter task. However, the study analyses: 3-back task, cued GoNoGo task, flanker task, and number-letter task. The study, however, discusses the importance and/or relevance of some and not others. Comment on the importance and/or relevance of some tests and not others.

-       The literature search is brief. The main executive functions (anticipation and development of attention, impulse control and self-regulation, mental flexibility and use of feedback, planning and organisation, etc.) could be developed.

o   Contreras-Osorio F, Guzmán-Guzmán IP, Cerda-Vega E, Chirosa-Ríos L, Ramírez-Campillo R, Campos-Jara C. Effects of the Type of Sports Practice on the Executive Functions of Schoolchildren. Int J Environ Res Public Health. 2022 Mar 24;19(7):3886. doi: 10.3390/ijerph19073886. PMID: 35409571; PMCID: PMC8998109.

o   Bryant AM, Kerr ZY, Walton SR, Barr WB, Guskiewicz KM, McCrea MA, Brett BL. Investigating the association between subjective and objective performance-based cognitive function among former collegiate football players. Clin Neuropsychol. 2022 Jun 7:1-22. doi: 10.1080/13854046.2022.2083021. Epub ahead of print. PMID: 35670306.

MATERIAL AND METHODS

Participants

-       Line: 70: “(Mage = 14.26 years, SD = 1.35 years). The average training age (mean years of experience playing soccer in a structured academy) was Mtage = 9.12 years (SD = 2.51 years).”. These values would go under "results".

-       Replace these values with: The study was a survey (study type: descriptive cross-sectional) among football players in a national youth football academy in (insert country), conducted between (include dates: month and year). The study subjects were (sex) between (e.g. 14 and 15 years). It is important to describe the setting, locations, and relevant dates, including periods of recruitment, exposure, follow-up, and data collection

Neuropsychological EF Tasks

-       Line 78: “Computerized neuropsychological tests with Inquisit Lab 6 (Millisecond Software LLC, Seattle, WA, USA) were performed to describe EFs on a 17-inch screen and a QWERTZ keyboard. Are all devices used validated?

Inclusion criteria

-       Eligibility criteria and sources and methods of selection of participants are not indicated. Give the eligibility criteria. Has there been any exclusion criteria?

It is recommended to make subsections organize the information such as: Study Design, Declarations: Ethics Approval and Consent to Participate, Eligibility Criteria and Outcome Measures.

RESULTS

-       It is recommended to add a first table with the socio-demographic characteristics of the sample.

-       It is recommended to add a second table showing the most relevant results of the research. For example, a table of significant and non-significant correlations between the variables under study. So at a first glance the reader can see the main findings.

-       In the "results" section, the results obtained are to be shown. Therefore, after adding the tables, the data obtained can be commented on.

-       Line 137: “Sensitivity analysis could be quantified with a value between 0.125 and 0.538 to identify low performance in neuropsychological tasks by the total score of self-reported EFs (BRIEF-SB inventory)”. This paragraph would be more appropriately added in methodology.

-       Add in this section information on exposures and potential confounding factors.

DISCUSSION

-       Provide a cautious overall interpretation of the results taking into account the objectives, multiplicity of analyses, results of similar studies, and other relevant evidence. Add more information from other studies and researchers in the area and compare them with your results.

-       Line174: “The athletes examined in the present study were retrieved from an elite academy in German”. This is already in methodology.

Limitations

-       There is no discussion of early ages and their important influence important influence on performance in tasks involving cognitive skills.

CONCLUSION

-       Future research directions may also be mentioned. Perhaps a questionnaire could also be designed in future research to adapt the type of responses.

REFERENCES

-       References follow the indicated style

Author Response

[Comment 1]:

The Abstract is properly structured, but according to the magazine's instructions (https://www.mdpi.com/journal/behavsci/instructions), what was the methodology used? (change the structure). For example, "68 young professional football players (Mage = 14.26 ± 1.35 years) from a national youth academy were included in the study. Relationships between neuropsychological tasks (3-back task, cue GoNoGo task, flanker task and number-letter task) and a self-report for examining EFs (BRIEF-SB)".

[Response]:

Thank you very much for your comment on the structure of the abstract. In my opinion, the abstract follows the instructions of the journal. To show this, I added the abstract with subpoints here:

Background: Cognitive diagnostics, especially the measurement of executive functions (EFs) in the context of sports and talent diagnostics, is a popular research topic. However, research is lacking on how self-reports are sufficient to examine the EFs of athletes for performance diagnostics. Methods: Thus, the current study aims to evaluate the relationships between neuropsychological tasks (3-back task, cued GoNoGo task, flanker task, and number-letter task) and a self-report for examining EFs (BRIEF-SB). Furthermore, it should be investigated whether it is possible to predict the outcome of EF tasks using a self-report inventory. Therefore, 68 young professional soccer players (Mage = 14.26 ±1.35 years) from a national youth academy were included in the study. Results: The weak-to-moderate correlations (r = .000, p = .999 to r = -.442, p < .01) and the results of sensitivity analysis (0.125 to 0.538) do not suggest using a self-report of EFs for cognitive performance diagnostics. Conclusion: The inventory is only suitable for identifying executive dysfunctions in athletes recovering from head injuries or concussions.

[Comment 2]:

The title or abstract should inform that the type of study.

[Response]:

Thank you again for screening the study in such a precise manner. I agree with this comment. I added "- A cross-sectional study "to the title to inform that it is a study with humans.

[Comment 3]:

Line 9: words are shown in red and crossed out, check sentence. Applicable to the rest of the document.

[Response]:

Thank you for that hint. The study had corrections after the first submission, and I let the MS-Word corrections in the text to inform that this was revised before submission. I changed the font color in the statistics section.

[Comment 4]:

Line 31: "The EF tests (i.e., flanker task, n-back task, and trail-making task) can describe the cognitive performance of athletes". It is recommended to add this sentence at the end of the paragraph. Remove "in conclusion".

[Response]:

Thank you very much for this suggestion. I agree with the statement that this sentence fits well at the end of the paragraph. I wanted to add a definition of neuropsychological at that point, and the sentence fits well with this definition. I revised the sentence on neuropsychological assessment at the end of the paragraph, and I hope that this suits your idea:

"Knowledge is lacking about using a self-reporting inventory to assess the EF of athletes. The current exploratory study aims to examine the correlations between the common used neuropsychological tasks (3-back task, cued GoNoGo task, flanker task, and number-letter task) and scores of BRIEF-SB to assess EFS in athletes "(lines 70-73)

[Comment 5]:

The following tests are discussed in the introduction: flanker task, n-back task, trail-making task, 3-back task, cued GoNoGo task and number-letter task. However, the study analyses: 3-back task, cued GoNoGo task, flanker task, and number-letter task. The study, however, discusses the importance and/or relevance of some and not others. Comment on the importance and/or relevance of some tests and not others.

[Response]:

Thank you very much for the comment. The tasks describe the core EFs commonly described in the literature and used in a sports context. I used the definition of Diamond (2012) to concentrate on inhibition, working memory, and cognitive flexibility. The tasks could examine the EF performance of the participants. I added a sentence at the end of "Neuropsychological EF Tasks": "The tasks are validated and used in several studies to assess EFs described above [1]." (lines 94-95)

[Comment 6]:

The literature search is brief. The main executive functions (anticipation and development of attention, impulse control and self-regulation, mental flexibility and use of feedback, planning and organisation, etc.) could be developed.

Contreras-Osorio F, Guzmán-Guzmán IP, Cerda-Vega E, Chirosa-Ríos L, Ramírez-Campillo R, Campos-Jara C. Effects of the Type of Sports Practice on the Executive Functions of Schoolchildren. Int J Environ Res Public Health. 2022 Mar 24;19(7):3886. doi: 10.3390/ijerph19073886. PMID: 35409571; PMCID: PMC8998109.

Bryant AM, Kerr ZY, Walton SR, Barr WB, Guskiewicz KM, McCrea MA, Brett BL. Investigating the association between subjective and objective performance-based cognitive function among former collegiate football players. Clin Neuropsychol. 2022 Jun 7:1-22. doi: 10.1080/13854046.2022.2083021. Epub ahead of print. PMID: 35670306.

[Response]:

Thank you very much for the comment and the suggestions for strengthening the background of the study. I agree with the statement and added a citation of the second suggested paper. However, the first one does not fit well in this context, and in the general background description, I cited a review focussing on this topic.

"The second study's findings of REF [] were that subjective cognition is not significantly associated with any objective measures of cognitive functioning in adults. "(lines 75-77)

REF:

Bryant AM, Kerr ZY, Walton SR, Barr WB, Guskiewicz KM, McCrea MA, Brett BL. Investigating the association between subjective and objective performance-based cognitive function among former collegiate football players. Clin Neuropsychol. 2022 Jun 7:1-22. doi: 10.1080/13854046.2022.2083021. Epub ahead of print. PMID: 35670306.

[Comment 7]:

Line: 70: "(Mage = 14.26 years, SD = 1.35 years). The average training age (mean years of experience playing soccer in a structured academy) was Mtage = 9.12 years (SD = 2.51 years).". These values would go under "results".

[Response]:

Thank you very much for the comment. I think it is legit to report the characteristics of the sample at this point. However, I think it wouldn't be nice to split the manuscript into too many subsections because it is a Brief Report. I hope it is fine with you.

[Comment 8]:

Replace these values with: The study was a survey (study type: descriptive cross-sectional) among football players in a national youth football academy in (insert country), conducted between (include dates: month and year).

The study subjects were (sex) between (e.g. 14 and 15 years). It is important to describe the setting, locations, and relevant dates, including periods of recruitment, exposure, follow-up, and data collection

[Response]:

Thank you very much. I added the following sentence on study type and substitutional information on the sex of the participants:

"The study was a survey (study type: descriptive cross-sectional) among professional youth soccer players. "(lines 117-118)

[Comment 9]:

Line 78: "Computerized neuropsychological tests with Inquisit Lab 6 (Millisecond Software LLC, Seattle, WA, USA) were performed to describe EFs on a 17-inch screen and a QWERTZ keyboard. Are all devices used validated?

[Response]:

The tasks and devices are validated. I added the following sentence: "The tasks are validated and used in several studies to assess EFs described above [1,4]. "(lines)

[Comment 10]:

Eligibility criteria and sources and methods of selection of participants are not indicated. Give the eligibility criteria. Has there been any exclusion criteria?

[Response]:

Thank you very much for this comment. I agree with that. I added a sentence on inclusion and exclusion criteria: "The inclusion criteria were an affiliation with the youth academy and a frequent training and physical health. Exclusion criteria were uncorrected ametropia or a concussion history for the last six months. "(lines 82-84)

[Comment 11]:

It is recommended to make subsections organize the information such as: Study Design, Declarations: Ethics Approval and Consent to Participate, Eligibility Criteria and Outcome Measures.

[Response]:

Thank you very much for your comment. I understand the reason for this suggestion. However, I think the subdivision of the manuscript is not necessary for a Brief Report.

[Comment 12]:

It is recommended to add a first table with the socio-demographic characteristics of the sample.

[Response]:

I have no further socio-demographic characteristics of the participants described in the Participants section.

[Comment 13]:

It is recommended to add a second table showing the most relevant results of the research. For example, a table of significant and non-significant correlations between the variables under study. So at a first glance the reader can see the main findings.

[Response]:

Thank you very much for the suggestion. I added the entire correlational analysis in the supplementary material. Unfortunately, the table does not fit in the text because it is too large. I hope that suits your idea.

[Comment 14]:

In the "results" section, the results obtained are to be shown. Therefore, after adding the tables, the data obtained can be commented on.

[Response]: You are right. Please see my response to comment 13.

[Comment 15]:

Line 137: "Sensitivity analysis could be quantified with a value between 0.125 and 0.538 to identify low performance in neuropsychological tasks by the total score of self-reported EFs (BRIEF-SB inventory)". This paragraph would be more appropriately added in methodology.

[Response]:

Thank you for your comment. I think this statement could be a result of a misunderstanding. The values are the results of sensitivity analysis. It was exploratory to identify the participants with the lowest performance in neuropsychological tasks using the self-report. I revised the sentence to make it more clear: "The sensitivity analysis results in values between 0.125 and 0.538 to identify low performance in neuropsychological tasks by using the total score of self-reported EFs (BRIEF-SB inventory). "(lines 157-159)

[Comment 16]:

Provide a cautious overall interpretation of the results taking into account the objectives, multiplicity of analyses, results of similar studies, and other relevant evidence. Add more information from other studies and researchers in the area and compare them with your results.

[Response]:

Thank you very much for the comment. I agree with you, and I revised the discussion and added two paragraphs to the suggested discussion:

"Even if the studies of REF [] and REF [] do not report findings regarding participants with the same characteristics, it could be stated that the studies show similar results. They also provide evidence that self-reports could not substitute neuropsychological measurements of EFs. In the current study, the self-report inventory (BRIEF-SB) is a valid instrument concerning participants' age. Still, the questionnaire asks for specific behavioral issues, trying to infer back to the EFs.

In most cases, athletes show no relevant issues concerning their EFs. Moreover, some studies show that it is difficult to differentiate between players in homogenous groups with high EF performance. The study shows that distinguishing between subjects is even more challenging when the examination method is inappropriate. "(lines 166-177)

[Comment 17]:

Line174: "The athletes examined in the present study were retrieved from an elite academy in German". This is already in methodology.

[Response]:

I agree with that statement. The cause for this duplicate is that at this point, it should be clear that these athletes build a homogenous high-performance group.

[Comment 18]:

Future research directions may also be mentioned. Perhaps a questionnaire could also be designed in future research to adapt the type of responses.

[Response]:

Thank you very much for the comment. I agree with the statement and added the following sentence:

"Future research in this field should focus on developing and validating specific inventories to examine EFs in athletes. The recently available inventories do not solve the problem and neglect special characteristics of athletes." (lines 204-207)

Round 2

Reviewer 1 Report

I recommend the article for publication

Reviewer 2 Report

Dear Author(s),

thank you very much for the revised version. I think the corrected and improved version is much better and appropriate for publishing. I do not have any further comments.

Best wishes!

Reviewer 3 Report

No further comments.